# Novel Elastic Threads for Intestinal Anastomoses: Feasibility and Mechanical Evaluation in a Porcine and Rabbit Model

**DOI:** 10.3390/ijms23105389

**Published:** 2022-05-11

**Authors:** Sophia M. Schmitz, Marius J. Helmedag, Klas-Moritz Kossel, Roman M. Eickhoff, Daniel Heise, Andreas Kroh, Mare Mechelinck, Thomas Gries, Stefan Jockenhoevel, Ulf P. Neumann, Andreas Lambertz

**Affiliations:** 1Department of General, Visceral- and Transplantation Surgery, RWTH Aachen University Hospital, Pauwelsstr. 30, 52074 Aachen, Germany; reickhoff@ukaachen.de (R.M.E.); dheise@ukaachen.de (D.H.); akroh@ukaachen.de (A.K.); uneumann@ukaachen.de (U.P.N.); alambertz@ukaachen.de (A.L.); 2Institute fuer Textiltechnik, RWTH Aachen University, 52074 Aachen, Germany; klas.kossel@rwth-aachen.de (K.-M.K.); thomas.gries@ita.rwth-aachen.de (T.G.); 3Department of Biohybrid and Medical Textiles (BioTex) at AME-Helmholtz Institute for Biomedical Engineering, RWTH Aachen University, 52074 Aachen, Germany; jockenhoevel@ame.rwth-aachen.de; 4Department of Anaesthesiology, RWTH Aachen University Hospital, Pauwelsstr. 30, 52074 Aachen, Germany; mmechelinck@ukaachen.de; 5Department of Surgery, Maastricht University Medical Center, P. Debyelaan 25, 6229 HX Maastricht, The Netherlands

**Keywords:** biomaterial, TPU, surgical thread, suture material, elastic suture, gastrointestinal anastomosis

## Abstract

Gastrointestinal anastomoses are an important source of postoperative complications. In particular, the ideal suturing material is still the subject of investigation. Therefore, this study aimed to evaluate a newly developed suturing material with elastic properties made from thermoplastic polyurethane (TPU); Polyvinylidene fluoride (PVDF) and TPU were tested in two different textures (round and a modified, “snowflake” structure) in 32 minipigs, with two anastomoses of the small intestine sutured 2 m apart. After 90 days, the anastomoses were evaluated for inflammation, the healing process, and foreign body reactions. A computer-assisted immunohistological analysis of staining for Ki67, CD68, smooth muscle actin (SMA), and Sirius red was performed using TissueFAXS. Additionally, the in vivo elastic properties of the material were assessed by measuring the suture tension in a rabbit model. Each suture was tested twice in three rabbits; No major surgical complications were observed and all anastomoses showed adequate wound healing. The Ki67+ count and SMA area differed between the groups (F (3, 66) = 5.884, *p* = 0.0013 and F (3, 56) = 6.880, *p* = 0.0005, respectively). In the TPU-snowflake material, the Ki67+ count was the lowest, while the SMA area provided the highest values. The CD68+ count and collagen I/III ratio did not differ between the groups (F (3, 69) = 2.646, *p* = 0.0558 and F (3, 54) = 0.496, *p* = 0.686, respectively). The suture tension measurements showed a significant reduction in suture tension loss for both the TPU threads; Suturing material made from TPU with elastic properties proved applicable for intestinal anastomoses in a porcine model. In addition, our results suggest a successful reduction in tissue incision and an overall suture tension homogenization.

## 1. Introduction

Gastrointestinal anastomoses provide a potential source of possible complications such as peritonitis, sepsis, and organ failure in abdominal surgery and are therefore a highly relevant issue in a surgeon’s craft [1,2]. Several risk factors for anastomotic leakage have been described. After rectal resection, for example, anastomoses close to the ano-cutaneous line, pre-operative radiation, intra-operative adverse events, male gender, smoking, and age have been declared risk factors [1,3,4]. Different techniques for the suturing and stapling of anastomoses have been evaluated, but the risk of insufficiencies remains a constant problem [3]. There have been efforts to maximize the tissue compatibility of suture materials through material composition and alteration of surface [4]. Among recent innovations are profiled (“snowflake”-shaped) sutures, which have proven superior to the standard (round) suture material in terms of foreign body reaction in a rodent model [4,5].

An established suturing material is polyvinylidene fluoride (PVDF), which has been proven to be a material with a superior biocompatibility [6]. PVDF has shown very promising results in comparison to established sutures in terms of foreign body reaction and collagen remodelling [6]. However, PVDF does not show any relevant longitudinal elasticity in tensions applied during surgery.

Human tissue, on the other hand, has an elastic modulus that leads to visible stretching even without pathologic incident. Elastic suturing material has the theoretical advantage of complying with changes in the sewn tissue. Furthermore, reducing the initial pressure applied to the tissue by knot tying could lead to improved wound healing, and an effective limitation of tensile forces could be achieved using highly elastic threads [7]. Thus, in the field of mesh implantation research, elastic fibres have recently been introduced with the aim of diminishing adverse tissue reactions [8,9,10]. Of these materials, thermoplastic polyurethane (TPU) has thus far proven the most suitable for medical applications, because it can be produced without solvents and shows a high biocompatibility [11,12,13]. TPU has been evaluated for many years in mesh production and vascular grafts [14,15,16]. In comparison to standard polypropylene (PP) sutures, the use of a TPU suture material leads to an improved foreign body reaction and a reduction in peak suture tension [11]. Furthermore, elastic TPU threads resulted in a reduced incision into the surrounding tissue in a rodent experiment whilst showing comparable biocompatibility to PVDF sutures [5]. Elastic polyurethane threads have already been successfully used for abdominal wall closure in rabbits, with favourable results [11,17,18].

However, suturing material with elastic properties has not been assessed for its usability in gastrointestinal anastomoses thus far. The aim of this study was to analyse the in vivo tension behaviour in a rabbit model and to evaluate the properties of circular- and “snowflake”-shaped TPU and PVDF sutures for the sewing of anastomoses of the small intestine in a porcine model.

## 2. Results

### 2.1. Minipig Surgery

All surgeries were carried out without any complications. All animals awoke immediately after the surgery and recovered quickly. A stepwise diet build-up was tolerated in all animals. None of the animals died in the postoperative observational period. During follow-up, there were no signs of wound infection, abscess formation, or abdominal wall hernia. There were no differences in weight at the time of euthanasia (F (3, 28) = 0.5706, *p* = 0.6391).

### 2.2. Macroscopic Evaluation

All anastomoses showed signs of good wound healing without any indication of leakage or abscess in the surrounding area. See Figure 1 for an example of an anastomosis at the time of explantation.

### 2.3. Microscopic Evaluation including Immunohistochemistry

Both modified snowflake threads showed intrafilamentous granuloma formation, which has been described previously [5]. There were no differences between the groups for inner and outer granuloma size, nor for the size of the comet-tail-like infiltrate (see Table 1).

Ki67 showed significant differences between the study groups (F (3, 66) = 5.884, *p* = 0.0013). A multiple comparisons test revealed a lower count of Ki67 nuclei in TPU-snowflake threads in comparison to PVDF threads of both the round and snowflake-shape varieties (*p*-values 0.0067 and 0.0062, respectively). See Figure 2 for exemplary histological Ki67 stains. The SMA area also showed a significant difference between the groups (F (3, 56) = 6, *p*-value 0.0005) with significantly larger areas in TPU-snowflake threads compared to all other groups (see Figure 3). The collagen I/III ratio did not show a significant difference between the groups (F (3, 54) = 0.4964, *p*-value = 0.6863) (see Figure 4). See Table 2 for an overview of the immunohistological staining.

### 2.4. Rabbit Suture Tension Measurements

No complications occurred during rabbits’ surgery until the finalization of the animals.

Both of the highly elastic TPU sutures significantly homogenized the mean suture tension range for both the round and snowflake configurations. This means that the average difference between the maximum applied suture tension at knotting and the remaining suture tension at steady state were lower in TPU threads compared to the round standard PVDF suture (TPU round *p* = 0.033, TPU snowflake *p* = 0.053) (see Figure 5 and Figure 6). The profiled PVDF suture did not significantly reduce the suture tension range (*p* = 0.998). The profiled TPU suture had to be knotted with a deliberate and consistent technique, as the abrupt rupture of the suture occurred on rare occasions during the experiment.

## 3. Discussion

The insufficiency of gastrointestinal anastomoses continues to be a clinically relevant complication in abdominal surgery and accounts for a high number of severe complications, especially in colorectal surgery [1,2,17,18,19,20]. In this study, we investigated the usability of elastic threads for the suturing of intestinal anastomoses to improve wound healing properties. TPU, which was used for elastic threads in this study, has shown favourable mechanical behaviour and excellent biocompatibility in previous in vivo and in vitro studies [11,12,13,14,15,16]. In this study, all the tested threads proved their usability for this application. Anastomotic leakage has been reported to be the most important outcome variable, and there were no clinical or pathological signs of anastomotic leakage for any of the tested sutures in this study [21]. As human and animal tissues have elastic properties, highly elastic sutures show an improved adaptability to the mended tissue due to a reduced loss of tension after knotting. Recently, France et al. reported on facilitated wound healing after treatment of commercially available suture material through viscoelastically-induced mechanotransduction [10]. Additionally, possible variances in the surgeon’s applied suture tension can be homogenized as demonstrated by the significantly reduced tension loss in the in vivo suture tension measurements. This is especially important considering that even advanced surgeons fail to replicate similar suture tensions [22]. In line with these results, Helmedag et al. showed a superior in vitro tension behaviour of TPU threads compared to PVDF, hallmarked by a reduced thread incision into surrogate tissue [5]. It could be argued that suture tension ranges can be determined by the elastic properties of the thread in combination with the suture diameter. The snowflake-like cross section did not improve the suture tension loss significantly for either material and had small handling deficiencies. Considering other applications for elastic threads, they have been shown to mimic the mechanical behaviour of healthy tissue in a rabbit model of laparotomy closure [23]. In another study by Lambertz et al., using TPU threads for closure of the abdominal wall in a rabbit model resulted in macroscopically equally adequate closure, while granuloma size was comparable [9]. The mechanical properties of TPU that are beneficial for laparotomy closure (e.g., elastic elongation in combination with sufficient tensile strength) appear to be equally beneficial for intestinal anastomoses. The rather long observational period of 90 days in this study should be considered as robust data to exclude later cases of anastomotic leakage or stricture caused by the novel sutures. Remarkably, in contrast to most other animal studies on anastomotic healing, in this study we provide additional insight into the histological process of anastomotic healing [21]. In this study, the collagen I/III ratio did not show a significant difference between the groups. In contrast, the collagen I/III ratio was significantly higher in fascia closure with TPU threads after 21 days in a study by Lambertz et al. [9]. Bellon et al. also found higher collagen I deposits in elastic polyurethane threats in rabbits after 3 weeks, suggesting a higher mechanical stability than the non-elastic threads [24]. Simon-Allue et al. provided similar results, albeit in a semi-quantitative manner; after 180 days, collagen I expression was not different to other thread materials and inferior to the control group [23]. It has to be taken into account that wound healing in gastrointestinal anastomosis is different from skin or fascia, and therefore the results might only be comparable to a limited extent. However, we did find a larger area of SMA in TPU threads. This might be a sign of superior healing and improved mechanical stability in the anastomoses sewn with TPU.

Another important parameter for implanted biomaterial is the foreign body reaction. The size of foreign body granuloma was comparable between the groups in our study, while Ki67 expression, as a marker for proliferation, was the lowest in the elastic snowflake suture group. This might indicate a more favourable lower foreign body reaction in the TPU snowflake group. This is in line with a recent study by Eickhoff et al., which also found a significantly lower foreign body reaction, indicated by lower Ki67 expression, in snowflake shaped sutures in a rodent model [4]. In contrast, Helmedag et al. showed a comparable foreign body reaction between TPU and PVDF threads [5].

This study has some limitations to be mentioned: as this was the first study to show the feasibility of intestinal small bowel anastomoses with elastic sutures in a porcine model, we did not evaluate anastomotic healing at different time points. Furthermore, there are scarce data on anastomotic healing in porcine models [21] that can be used to estimate expected complications. For example, the small bowel anastomoses evaluated in this study showed good healing tendencies in general, especially compared to other anastomoses such as colorectal anastomoses which are prone to insufficiency. Small differences in healing behaviour and especially differences in anastomotic leakage rate could therefore be easily missed due to lacking statistical power. Furthermore, image-related histological analysis can only provide an estimation of the true cellular distribution. This factor was minimized in this study by using the image analysis software StrataQuest but remains to be noticed.

Our findings have manifold clinical impacts: this is the first study to demonstrate the potential of highly elastic suturing material (TPU) for intestinal anastomoses. Additionally, TPU threads have already been shown to reduce tissue incision, which is beneficial for wound healing, while showing a comparable biocompatibility [5]. The novel suture material proved highly applicable in the sewing of intestinal anastomoses with signs towards an improved tissue reaction compared to standard suture material. This is an entirely novel and hardly explored approach to improve the healing process and therefore safety of intestinal anastomoses. Therefore, the potential of TPU sutures should be further explored in more critical intestinal anastomoses, such as the colon or pancreas.

## 4. Materials and Methods

The development of the newly established sutures has been reported previously [5]. All suture materials were manufactured by the Institut fuer Textiltechnik (ITA), RWTH Aachen, Aachen, Germany in collaboration with the Department of Biohybrid and Medical Textiles (BioTex) at the AME-Helmholtz Institute for Biomedical Engineering, RWTH Aachen University, Aachen, Germany. The used materials were PVDF and TPU. Both materials were tested in round and profiled (snowflake) shape as described previously with a comparable cross-sectional area corresponding to the United States Pharmacopeia (USP) 4-0 [4,5]. Briefly, TPU and PVDF monofilaments were extruded with a melt spinning process at 255 °C (PVDF) and 218 °C (TPU) and subsequent cooling in a water bath. The resulting filaments were then wound upon spools for usage.

### 4.1. Animals

All animal experiments were performed at the Central Animal Facilities of the University Hospital of RWTH Aachen University. The experiments were approved by the Governmental Animal Care and Use Committee (LANUV, Landesamt für Natur, Umwelt und Verbaucherschutz Nordrhein-Westfalen, Recklinghausen, Germany; reference AZ 81-02.04.2018.A152). Three New Zealand White rabbits with a body weight (BW) of approximately 3000 g were used for the in vivo suture tension experiments. Thirty-two mini-pigs (Sus scrofa domesticus) weighing 40–60 kg were randomly divided into four groups (*n* = 8 in each group). All animals were kept under constant, standardized conditions with free access to water (ad libitum) and food (2–3% of BW per day) and a constant temperature for two weeks prior to surgery. Each of the four groups was assigned a different suturing material: (1) PVDF round, (2) TPU round, (3) PVDF snowflake, or (4) TPU snowflake shape.

### 4.2. Surgical Procedures

#### 4.2.1. Minipig Small Bowel Anastomoses

Each animal was assigned one of the suture materials randomly. Surgical procedures were performed under general anaesthesia. The weight of each animal was determined prior to surgery. After the induction of anaesthesia, antibiotic prophylaxis with cefuroxime 750 mg and metronidazole 250 mg was administered intravenously. The animals were placed in a supine position and the abdominal skin was shaved and disinfected. Next, a midline minimal abdominal incision of 8–10 cm was performed. The small bowel was pulled out of the abdominal cavity and two sections (2 m apart) for sewing of an intestinal anastomosis were identified. The small bowel was divided at the identified sections and then sewn together straight and tension free with a running suture. The small bowel was then placed back into the abdominal cavity whilst avoiding any twisting or strangulation. Subsequently, the abdomen was closed in two layers, first the peritoneum and then the muscular fascia with Vicryl 0. The skin was closed using single-knot sutures (Premilene 2-0). All surgical procedures were performed by the same surgeon (S.M.S.). See Figure 7 for an image of the operative setup and Figure 8 for an image of a sewn small bowel anastomosis. After the surgery, the animals were observed by the surgeon until complete recovery. Postoperative analgesia was administered with Finadyne injected intramuscularly (1–2 mg/kg bodyweight, one dose every 24 h) for the first three days postoperatively. In the case of any signs of pain, analgesia was continued. Throughout the study period, the animals were carefully monitored for signs of wound healing disorders, seroma formation, or gastrointestinal problems. The animals were sacrificed with an intravenous injection of pentobarbital 90 days after the initial surgery, and the anastomoses were explanted and prepared for histological analysis.

#### 4.2.2. Histological Assessment

After explantation of the anastomoses, the specimens were fixed in formaldehyde. See Figure 3 for an example of an anastomosis at the time of explantation. The specimens were embedded in paraffin and cut in 3 µm (for immunohistochemistry) and 5 µm (for Sirius red and haematoxylin and eosin (H&E)) sections. All sections provided a longitudinal cut through the anastomoses. H&E staining was performed for all specimens.

The granuloma size was determined separately for the inner and outer granuloma at two different regions, and the size of the comet-tail-like infiltrate was measured according to Klink et al. [25].

For assessment of the foreign body reaction and healing process, immunohistochemical staining for CD68 (macrophage activity), Ki67 (proliferation), and smooth muscle actin (SMA) was performed according to the manufacturers’ instructions. For all immunohistochemical staining, visualization was performed with the Zytochem-Plus AP Polymer-Kit (Zytomed Systems). Macrophages (CD68) were identified by a 1:50 mouse monoclonal antibody from Dako, with the fixed specimen pretreated 3 times with microwave and citrate buffer (pH 6). Ki67 expression was investigated with a mouse monoclonal antibody MIB-1, 1:10 from Dako. For SMA, a 1:50 mouse monoclonal antibody (Dako) was used.

Sirius red staining was used for the determination of the collagen type I and type III fractions as described previously [26]. Sections of 5 µm were stained for 1 h in Picrosirius solution (0.1% solution of Sirius red F3Ba) in saturated aqueous piric acid. Sections were then washed, dehydrated, cleared, and mounted in resin. Examination of the sections was performed under a cross-polarization microscope. Images were acquired for all stainings with a Baumer Optronic HXG40c camera under 200-fold magnification.

The suture–tissue interface region was determined and analysis was performed for five suture–tissue interface regions in each sample. For microscopic imaging, a TissueFAXS Plus upright brightfield system (Tissuegnostics GmbH, Vienna, Austria) was used, and the analysis of positive cells was determined with the affiliated software (StrataQuest, Tissuegnostics GmbH, Vienna, Austria) in a semi-automatic fashion (see Figure 9). For analysis, the regions of interest (ROI) were set in an identical manner for each suture material. Proliferation staining was performed by determining the number of Ki67-positive nuclei with an identical threshold for each slide (see Figure 5). To quantify the smooth muscle cells, the total SMA-positive area of the stained tissue was similarly calculated separately for each slide (see Figure 6). Collagen fractions were obtained by analysis of the amount area of green- and red-orange-coloured collagen (see Figure 7). Blinding was performed for each specimen regarding the polymer used, while blinding regarding the suture structure was not possible due to the visibly different suture structure.

#### 4.2.3. Rabbit Small Bowel Suture Tension Measurements

General anaesthesia was induced by a subcutaneous injection of 0.1 mg/kg BW of medetomidine combined with 0.2 mL/kg BW 10% ketamine. Balanced anaesthesia was maintained by the intravenous infusion of 0.01 mg/kg BW/h fentanyl and 2 vol. % isoflurane after intubation. The rabbits were placed in a supine position, and the abdominal area was shaved and disinfected. A midline incision was performed. Eight suture loops were placed evenly distributed in the small bowel of each rabbit. All four suture groups were tested twice in each rabbit, so a total of six measurements were performed for each thread. Suture tension measurements were performed by custom-modified miniature bending beam load cells optimized for in vivo use and connected to a multichannel A/D USB Interface (both Burster Präzisionsmeßtechnik, Gersbach, Germany). The modification included a baseplate as an abutment for the sutured small intestine and a V-shaped suture guide designed to transfer the suture tension to the load cell (see Figure 10). Four load cells were connected to the A/D interface to enable parallel measurements: four measurements were performed in parallel (see Figure 11). The suture tension behaviour of each suture was continuously recorded from suture knotting until the end of the measurement after 15 min according to previous experiments by Klink et al. [7]. Further mechanical properties of the used sutures have been published previously [5]. All sutures achieved a steady state of suture tension within this time. The tension diminishment was calculated as the difference between the peak suture tension at the time of knotting and the tension minimum at steady state. The rabbits were euthanized by an intravenous injection of 400 mg/kg BW pentobarbital. All surgeries were performed by the same surgeon (M.H.).

### 4.3. Statistical Analysis

Outliers ± 2 standard deviations (SD) from the mean were excluded from further analysis. Differences between groups were analysed by nonparametric one-way-ANOVA. Post-hoc analysis was performed by Tukey’s multiple comparisons test. A *p*-value < 0.05 was considered to be statistically significant. All analyses were carried out using Microsoft SPSS v18 and GraphPad Prism v8. Data are represented as the mean ± SD, unless otherwise indicated.

## Figures and Tables

**Figure 1 ijms-23-05389-f001:**
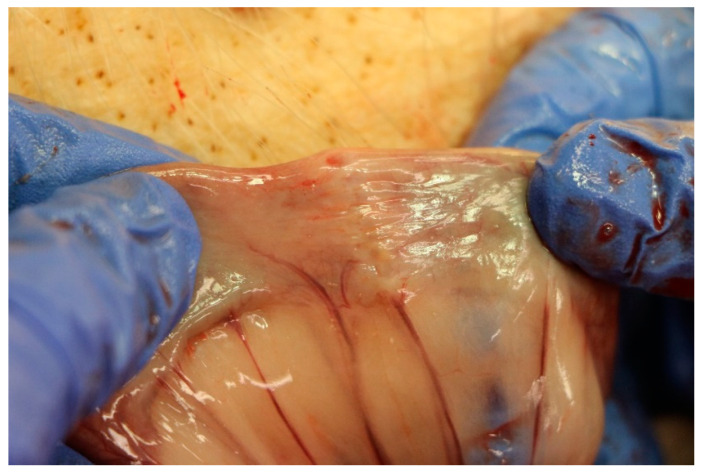
Example of the macroscopic impression of the anastomosis at the time of explantation.

**Figure 2 ijms-23-05389-f002:**
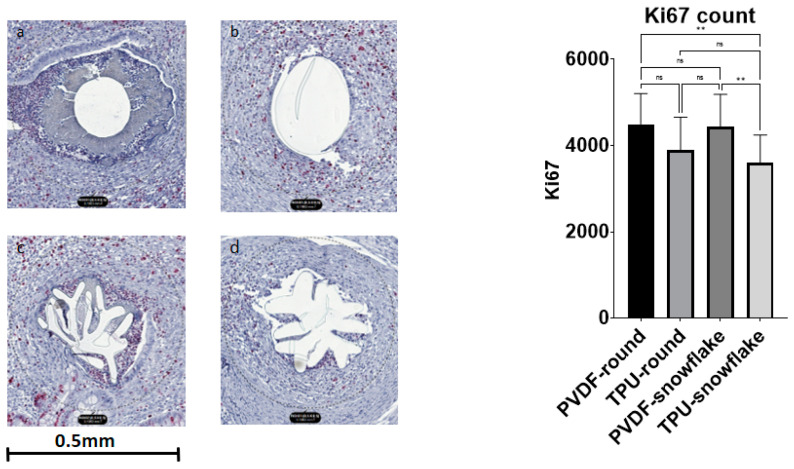
Immunohistological stainings for Ki67. (**a**) PVDF-round, (**b**) TPU-round, (**c**) PVDF-snowflake, (**d**) TPU-snowflake. There was a significant difference between the groups (F (3, 66) = 5.884, *p* = 0.0013). Abbreviations: ns: not significant, PVDF: polyvinylidene fluoride, TPU: thermoplastic polyurethane, ** indicates *p* < 0.01.

**Figure 3 ijms-23-05389-f003:**
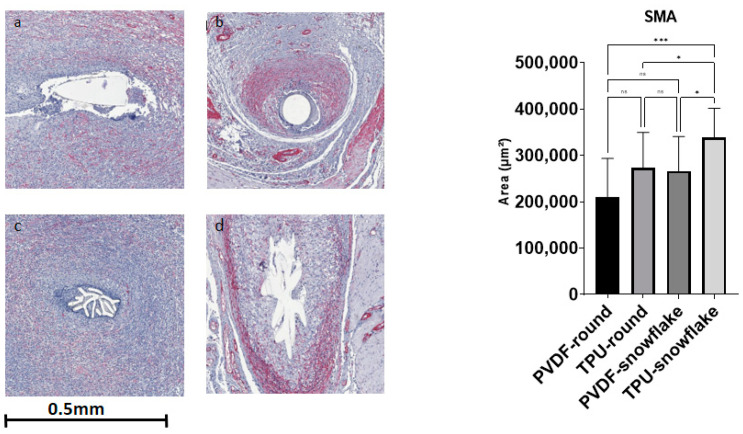
Immunohistological stainings for SMA. (**a**) PVDF-round, (**b**) TPU-round, (**c**) PVDF-snowflake, (**d**) TPU-snowflake. There was a significant difference between the groups (F (3, 66) = 5.884, *p* = 0.0013). Abbreviations: ns: not significant, PVDF: polyvinylidene fluoride, TPU: thermoplastic polyurethane, SMA: smooth muscle actin; * indicates *p* < 0.05, *** indicates *p* < 0.001.

**Figure 4 ijms-23-05389-f004:**
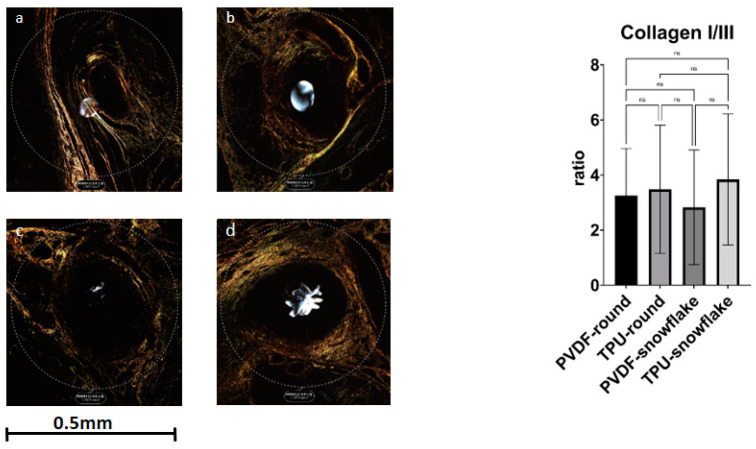
Cross-polarization analysis for Sirius red staining. (**a**) PVDF-round, (**b**) TPU-round, (**c**) PVDF-snowflake, (**d**) TPU-snowflake. There was no significant difference between the groups (F (3, 54) = 0.4964, *p*-value = 0.6863). Abbreviations: ns: not significant.

**Figure 5 ijms-23-05389-f005:**
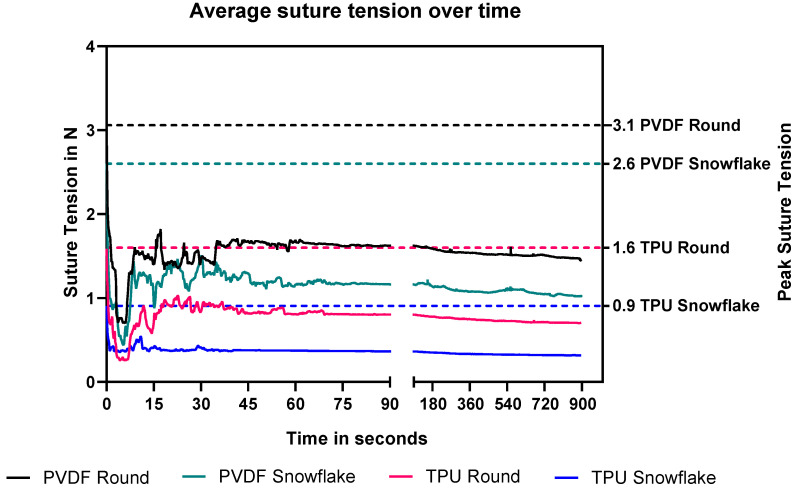
Average suture tension over time. Note the varying scale of the *x*-axis to better illuminate the short-term and long-term tension behaviour of the sutures. Second *y*-axis highlights peak suture tensions of each suture.

**Figure 6 ijms-23-05389-f006:**
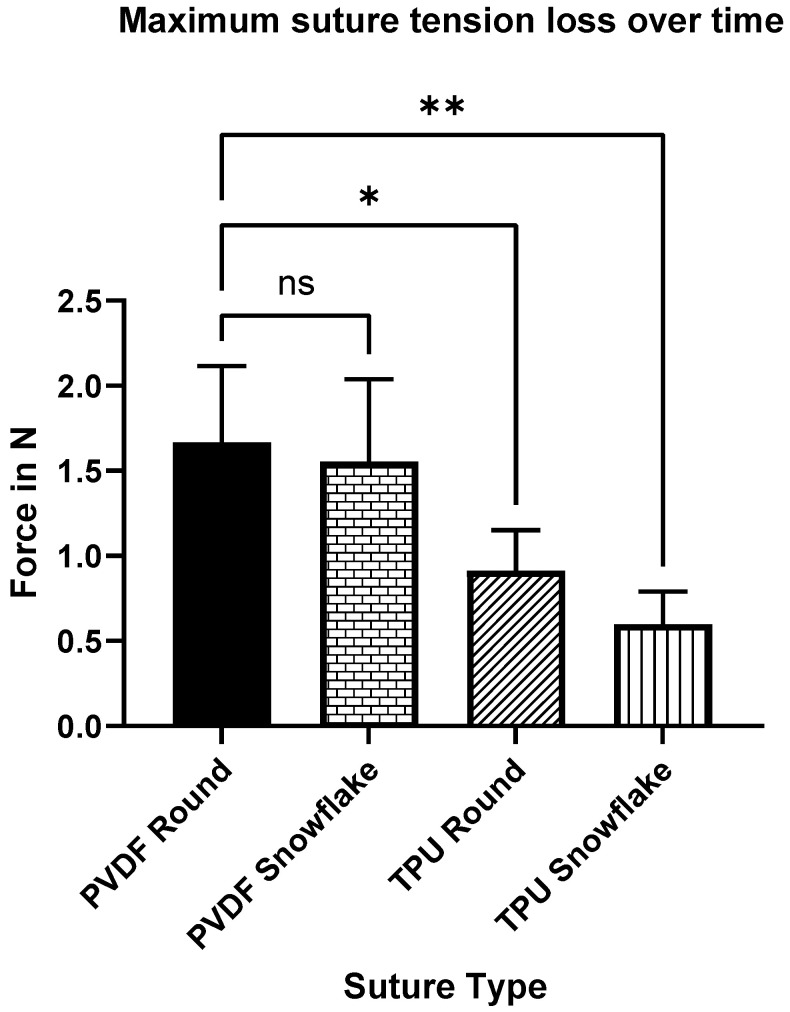
Maximum tension difference between peak suture tension and suture tension steady state after 15 min. Both TPU threads showed a significantly reduced tension loss vs. round PVDF sutures. Significance levels: *p* < 0.05 = *; *p* < 0.01 = **. Abbreviations: ns: not significant.

**Figure 7 ijms-23-05389-f007:**
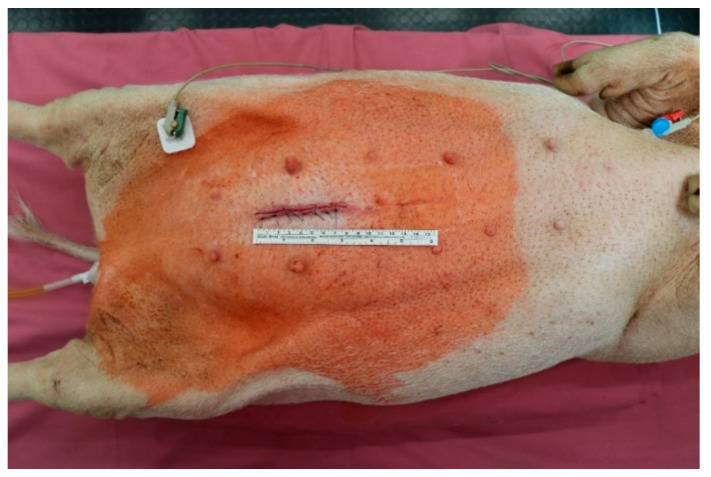
Operative setup after closure of the abdominal wall. The animal was placed in a supine position. The picture shows the mini-laparotomy of 7 cm length.

**Figure 8 ijms-23-05389-f008:**
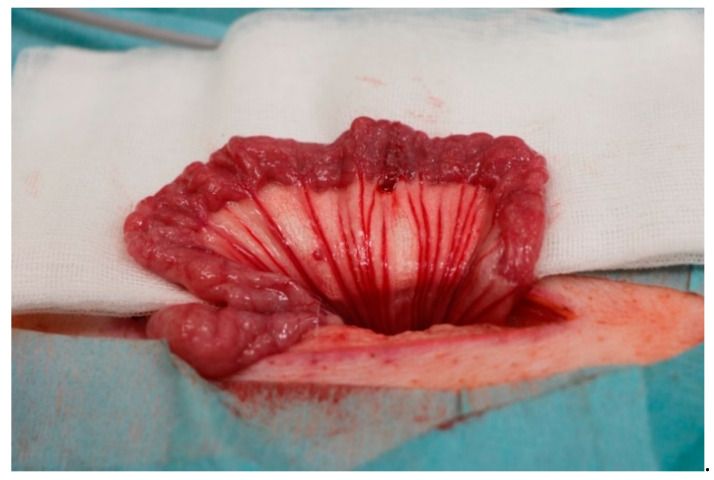
Operative setup after sewing of the first intestinal anastomosis. The small bowel was placed in front of the abdominal wall, and the anastomosis was sewn with running sutures. The procedure was equal for all of the used thread materials.

**Figure 9 ijms-23-05389-f009:**
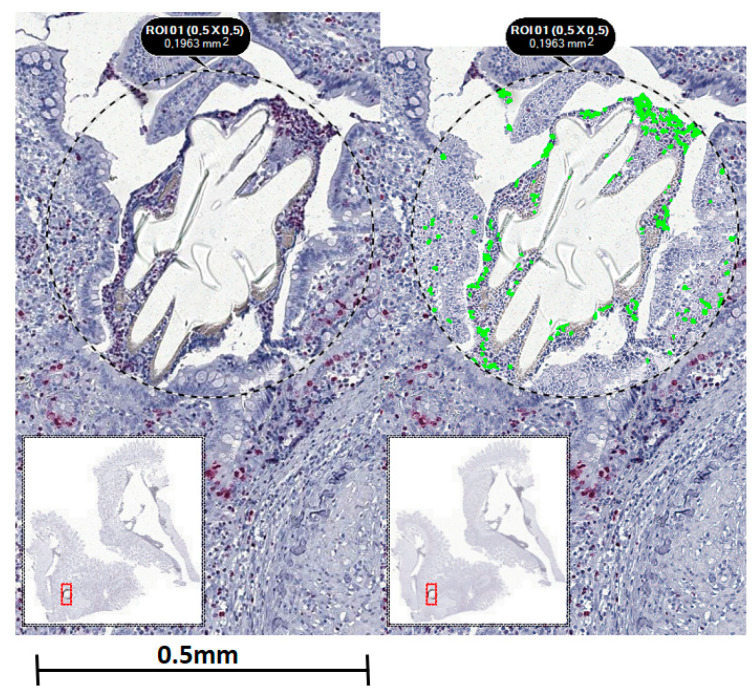
Example of semi-automatic analysis of the suture-tissue interface. Standard regions of interest (ROI) were set around each suture manually and then analysed automatically using StrataQuest software.

**Figure 10 ijms-23-05389-f010:**
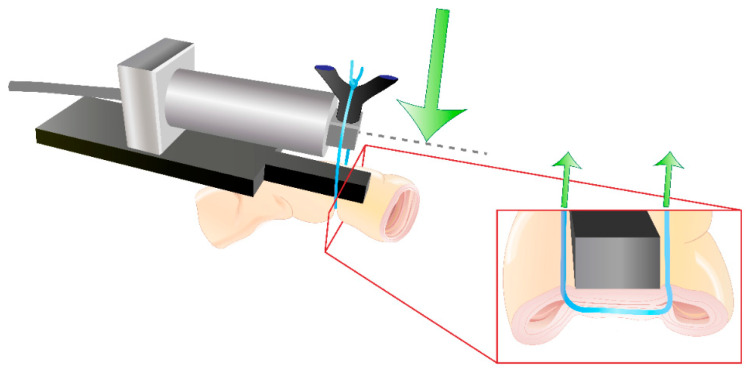
Visualization of the custom modified suture tension load cell for the in vivo tension measurements in rabbits. Green arrows indicate the measured suture tension forces.

**Figure 11 ijms-23-05389-f011:**
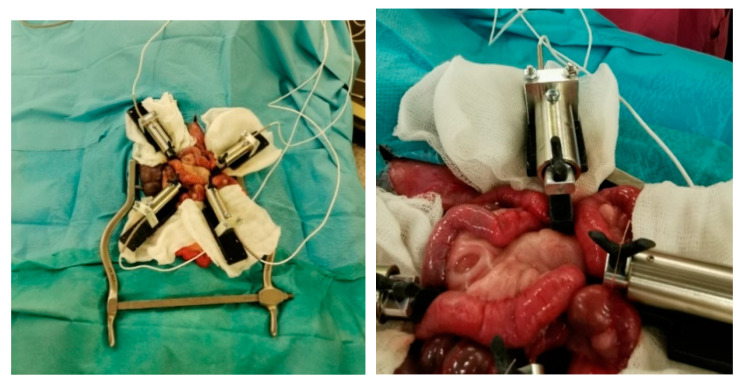
Experimental setup for parallel suture tension measurements.

**Table 1 ijms-23-05389-t001:** Microscopic characteristics of the different threads. There were no statistically significant differences between the tested materials in either granuloma size or comet-tail infiltrate. Abbreviations: PVDF: polyvinylidene fluoride, SD: standard derivation, TPU: thermoplastic polyurethane. Data are represented as the mean (SD).

	PVDF-Round	TPU-Round	PVDF-Snowflake	TPU-Snowflake	*p*-Value
Inner Granuloma	30.8 (15.1)	40.6 (25)	29.5 (5.6)	33.4 (14.2)	0.3711
Outer Granuloma	97.1 (45.6)	118.4 (71.8)	100.3 (34.4)	113.5 (41.9)	0.7141
Comet-Tail Infiltrate	395.8 (123.9)	357.7 (160.1)	340.2 (98.6)	400.9 (117.5)	0.6187

**Table 2 ijms-23-05389-t002:** Immunohistochemical characteristics of the different threads. The Ki67 count and SMA area showed significant differences between the groups, while the collagen I/III ratio and CD68 count were not different between the groups. Abbreviations: PVDF: polyvinylidene fluoride, TPU: thermoplastic polyurethane, SD: standard derivation, SMA: smooth muscle actin. Data are represented as the mean (SD).

	PVDF-Round	TPU-Round	PVDF-Snowflake	TPU-Snowflake	*p*-Value
Ki67 [nuclei count]	4497 (711)	3903 (756)	4438 (749)	3610 (637)	0.0013
SMA [area, µm^2^]	210,784 (82,404)	272,770 (76,473)	265,719 (74,756)	337,952 (63,691)	0.0005
Collagen I/III ratio	3.3 (1.7)	3.5 (2.4)	2.8 (0.2)	3.8 (2.4)	0.6863
CD68 [nuclei count]	3531 (1243)	3346(886)	2978 (760)	2833.12 (531)	0.0558

## Data Availability

The data presented in this study are available on request from the corresponding author.

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
