# Peer review of "Novel Elastic Threads for Intestinal Anastomoses: Feasibility and Mechanical Evaluation in a Porcine and Rabbit Model"

_ijms, 2022, doi:10.3390/ijms23105389_

Round 1

Reviewer 1 Report

Novel elastic threads for intestinal anastomoses: feasibility and mechanical evaluation in a porcine and rabbit model

  • This study is aimed to evaluate a newly developed suturing material made from thermoplastic polyurethane (TPU) and Polyvinylidene Fluoride (PVDF).
  • PVDF and TPU were tested in 2 different textures (round and modified, “snowflake” structure) in 32 Minipigs, with two anastomoses of the small intestine sutured 2m apart. After 90 days, the anastomoses were evaluated for inflammation, the healing process, and foreign body reaction.
  • Immunohistological staining for Ki67, CD68, Smooth muscle actin (SMA), and Sirius Red was performed using TissueFaxs. In-vivo elastic properties were assessed by measuring suture tension in a rabbit model.
  • Results show
  • No major surgical complications and all anastomoses showed adequate wound healing. Ki67+ count and SMA area differed between groups.
  • In TPU-snowflake material Ki67+ count was the lowest, while the SMA area provided the highest values.
  • CD68+ count and Collagen I / III ratio did not differ between groups.
  • Suture tension measurements showed a significant reduction in suture tension loss for both TPU threads;

Technical issues:

  1. TPU threads show significantly reduced tension loss in Fig. 10, But these tensions are all within 2 N which is a very level of forces. Even though the difference between TPU and others is statistically significant, the practical difference could be very minimal. There must be alternative ways to verify the practical effects of these low levels of tension.
  2. I understand the difficulties in vivo tests, but I think it may not be appropriate to call the “steady state” at ~15 minutes (~900 s). Because the elasticity of suture threads is not measured in-vitro by standard methods in this study. Scientifically, the elasticity of materials is rigorously defined. There usually could be viscoelastic behavior in polymeric materials at different time scales. Please also note that TPU has certain plastic or non-elastic behavior that can’t be ignored. In other words, the mechanical assessment in this study is very limited to in-vivo measurements.
  3. The expression of Ki67 is widely used in the routine pathological investigation as a proliferation marker. Fig. 4 and Table 2 show a significant difference between groups in the immunohistological staining of Ki67. These are results based on the statistical numbers. However, these numbers are from the image analysis which is only a qualitative assessment. It’s difficult to know how images correlate with the true quantities of Ki67 in tissues. In fact, by looking at Table 2, a quite large standard deviations-to-mean ratio (PVDF-round (15.81%) TPU-round (19.36%) PVDF-snowflake (16.87%) TPU-snowflake (17.65%)) in all cases could compromise the conclusion. Similar issues exist for the collagen I/III ratios.
  4. It may be necessary to have some biochemical analysis to support the immunohistological staining. The biochemical analysis can provide better quantitative validation or confirmation of the postsurgical results.
  5. I did not find numbers from the measurement of CD68+. They should be included in the report inappropriate sections, tables, or figures as a credential for readers.

Reviewer 2 Report

The presented results can contribute into development of mechanobiology and have potential to be used in clinics.

The aim formulated at the end of Introduction mu must be clearly justified by providing more available data about the role of tension. The conclusions must be presented not only in the abstract but mainly in discussion. It would be beneficial to present future perspectives demonstrating novelty and originality of obtained results.

Author Response

Dear Reviewer,

thank you very much for your appreciation of our manuscript. We added further evidence to the discussion section and emphasized the meaning of our results.

We would furthermore like to mention a copy-issue that was solved. Section 2.1. Minipig surgery appeared twice in the submission, while 2.2. Macroscopic evaluation was missing. We apologize for the inconvenience.

Thank you again for your valuable comments.

Round 2

Reviewer 1 Report

The authors replied to all-important enquires from the 1st round of review. Although there are some missing points and room for improvement in this report, I think it can be accepted for publication in its current content.

Please check typo errors and spelling before transferring the manuscripts to a publishable form.